



# Effects of clearfell harvesting on soil CO₂, CH₄ and N₂O fluxes in an upland Sitka spruce stand in England

Sirwan Yamulki[1], Jack Forster[1], Georgios Xenakis[2], Adam Ash[2], Jacqui Brunt[1], Mike Perks[2] and James I. L. Morison[1]

[1]Forest Research, Centre for Sustainable Forestry and Climate Change, Alice Holt Lodge, Farnham, Surrey, GU10 4LH, UK
[2]Forest Research, Centre for Sustainable Forestry and Climate Change, Roslin, Midlothian, EH25 9SY, UK.

*Correspondence to*: Sirwan Yamulki (sirwan.yamulki@forestresearch.gov.uk)

**Abstract.** The effect of clearfell harvesting on soil greenhouse gas (GHG) fluxes of carbon dioxide ($CO_2$), methane ($CH_4$) and nitrous oxide ($N_2O$) was assessed in a Sitka spruce forest growing on a peaty gley organo-mineral soil in northern England. Fluxes from the soil and litter layer were measured monthly by the closed chamber method and gas chromatography over four years in two mature stands, with one area harvested after the first year. Concurrent measurements of soil temperature and moisture helped to elucidate reasons for the changes in fluxes. In the three years after felling, there was a significant increase in the soil temperature, particularly between June and November (3 to 5 ℃ higher), and in soil moisture which was 62% higher in the felled area, and these had pronounced effects on the GHG balance, in addition to the removal of the trees and their carbon input to the soil. Annual soil $CO_2$ effluxes reduced to almost a third in the first year after felling (a drop from 24.0 to 8.9 t $CO_2$ ha⁻¹ yr⁻¹) and half in the second and third year (mean 11.8 t $CO_2$ ha⁻¹ yr⁻¹) compared to before felling, while those from the unfelled area were little changed. Annual effluxes of $N_2O$ more than doubled in the first two years (from 1.0 to 2.3 and 2.5 t $CO_2$e ha⁻¹ yr⁻¹, respectively), although by the third year they were only 20% higher (1.2 t $CO_2$e ha⁻¹ yr⁻¹). $CH_4$ fluxes changed from a small net uptake of -0.03 t $CO_2$e ha⁻¹ yr⁻¹ before felling to a small efflux increasing over the 3 years to 0.34 t $CO_2$e ha⁻¹ yr⁻¹, presumably because of the wetter soil after felling.

Soil $CO_2$ effluxes dominated the total net GHG emission calculated using the global warming potential (GWP) of the three gases, but $N_2O$ contributed up to 20% of the total annual emissions. This study showed fluxes of $CO_2$, $CH_4$ and $N_2O$ responded differently to clearfelling due to the significant changes in soil biotic and abiotic factors and showed large variations between years. This demonstrates the need for multi-year measurements of all GHGs to enable a robust estimate of the effect of the clearfell phase on the GHG balance of managed forests. This is one of a very few multi-year monitoring studies to assess the effect of clearfell harvesting on soil GHG fluxes.

## 1 Introduction

Forests cover approximately 30% (4.03 billion ha) of the earth's land surface and play a major role in cycling of soil carbon and greenhouse gases (GHG) (Le Quéré et al., 2015). Afforestation and forest management can contribute to GHG net emission mitigation aims by increasing the land-based carbon sink (Grassi et al., 2017). In the UK, 3.2 million ha are covered by forests, 13.2% of the land area (Forestry Commission, 2020), a substantial proportion of which have been planted over the last 60 to 100 years on peat and peaty gley upland soils (Smith et al., 2018). The UK government has pledged to reach 'net zero' emissions of GHGs by 2050 and to contribute to this goal, the Committee on Climate Change (2019) has recommended afforestation targets of more than 30,000 ha per year. Forest harvesting is an important activity as part of normal forestry practice but also for conversion to other land uses such as part of peatland restoration programmes. Approximately 20,000 ha of trees are felled annually in Great Britain for timber and other harvested wood products (Forestry Commission, 2016).

Forest clearfell harvesting is the phase of the forest management cycle which produces the most disturbance and therefore it is important that it does not impair long-term productivity and benefits. Clearfelling alters (typically increases) many soil factors that influence GHG fluxes. The physical factors include: soil water content and water table height owing to the absence of evapotranspiration from trees (Zerva and Mencuccini, 2005; Wu et al., 2011; Kulmala et al., 2014; Sundqvist



et al., 2014; Korkiakoski et al., 2019); soil temperature through reduced shading (Zerva and Mencuccini, 2005; Wu et al., 2011; Kulmala et al., 2014); soil bulk density, due to soil disturbance and compaction caused particularly by mechanised harvesting equipment (Yoshiro et al., 2008; Mojeremane et al., 2012). Chemical factors include: soil pH (Smolander et al., 1998; Kim, 2008; Kulmala et al., 2014; Sundqvist et al., 2014) and inputs of soil N and organic C (Smolander et al., 2015;

Tate et al., 2006; Hyvönen et al., 2012; Kulmala et al., 2014; Sundqvist et al., 2014), due to nutrient release from the decomposition of residual organic matter and root biomass. Interactions between these factors, particularly with the loss of plant litter input and root activity after felling, will strongly affect soil biological processes responsible for production and consumption of GHGs. They will particularly affect nitrous oxide ($N_2O$) production from aerobic nitrification and anaerobic denitrification, methane ($CH_4$) production by methanogenic organisms from anaerobic decomposition in oxygen-poor

environments or uptake by methanotrophs through oxidation in aerated soils, and carbon dioxide ($CO_2$) efflux during respiration and decomposition and uptake during photosynthesis. The cessation of the autotrophic respiration (Ra) component of the total soil respiration (Rt) after felling should cause a large decline in $CO_2$ efflux as a meta-analysis of soil respiration partitioning studies reported that the Ra/Rt ratio in temperate forest soils ranges from 20 to 59% (Subke et al., 2006). In addition the death of tree roots after felling will inhibit the microbial decomposition of root exudates reducing the $CO_2$ effluxes further.

Conversely, the increased soil temperatures following tree removal may increase soil heterotrophic respiration (Rh) (Yashiro et al., 2008).

There are a wide range of results published on the effects of forest clearfelling on soil GHG fluxes. For $CO_2$, the overall soil efflux after felling depends on the balance between the decrease in root respiration and the increase in microbial respiration in response to warmer soil temperatures and nutrient availability from decomposition of litter and other plant

materials such as brash. Therefore, some studies have shown $CO_2$ effluxes were reduced (Striegl and Wickland, 1998; Laporte et al., 2003; Zerva and Mencuccini, 2005; Goutal et al., 2012, Korkiakoski, 2019), increased (Tate et al., 2006; Kim, 2008; Kulmala et al., 2014), or unchanged (Toland and Zak, 1994; Butnor et al., 2006; Takakai et al., 2008; Yoshiro et al., 2008). The differences between studies on the effect of harvesting disturbance on GHG fluxes have also been attributed to microsite distribution within the harvested area (Laporte et al., 2003). According to Lavoie et al. (2013), the impact is often site specific,

affected by the severity of the disturbance or removal of surface organic matter (Fleming et al., 2006; Moroni et al., 2007; Slesak et al., 2010) or by length of time following harvest (Humphreys et al., 2006; Peng and Thomas, 2006; Olayuyigbe et al., 2012). Goutal et al. (2012) examined the duration of physical, chemical and biological disturbances in the soil following mechanized harvesting of a beech/oak forest in NE France. Their measurements showed that soil $CO_2$ effluxes reduced which they attributed to an increase in the frequency and duration of anoxic conditions resulting from poor soil gas diffusion after

heavy forestry traffic. Similarly, Kulmala et al. (2014) observed a slight decrease in soil $CO_2$ efflux in the first growing season after clear-cutting of a boreal Norway spruce stand, although this was probably due to decreased tree root respiration. However, during the following two years, $CO_2$ efflux at their clear-cut site was significantly higher than in their mature stand, which was attributed to increased decomposition, stimulated by higher soil moisture and temperature. They observed no significant difference in $CH_4$ uptake due to clear-cutting. However, others have shown that after harvesting forest soils turned

from a $CH_4$ sink to a source (Zerva and Mencuccini, 2005; Castro et al., 2000; Takakai et al., 2008; Yashiro et al., 2008; Sundqvist et al., 2014; Korkiakoski, 2019), showed reduced $CH_4$ uptake (Bradford et al., 2000; Wu et al., 2011; Yoshiro et al., 2008), showed no change (Kulmala et al., 2014) or even increased $CH_4$ uptake (Lavoie et al., 2013). For $N_2O$, some studies showed increased fluxes (Zerva and Mencuccini, 2005;Yoshiro et al., 2008; Takakai et al., 2008; Ullah et al., 2009; Korkiakoski, 2019), or had no clear change (Tate et al., 2006; Lavoie et al., 2013).

There is therefore little consistent information with which to predict the rate and duration of changes in the GHG balance by clearfelling with very few multiple-year studies (Wu et al., 2011). However, there is an urgent need to understand and quantify with long-term measurements the effect of forest clearfelling on the GHG budget and to incorporate these into life cycle analyses (Skiba et al., 2012). Therefore, the objective of this study was to conduct a relatively long-term (4 year)





assessment of the effect of clearfell harvesting on soil (including litter layer) fluxes of $CO_2$, $CH_4$ and $N_2O$ in a spruce plantation

on organo-mineral soil that is typical of many British upland forests.

## 2 Materials and methods

### 2.1 Site description and study layout

The study site was in Harwood Forest, Northumberland, north-east England, and comprised a forest area of approximately

4000 ha with an elevation of 200 to 400 m above sea level (Fig. 1). The regional climate is temperate oceanic, with a mean

annual rainfall of 1472 mm and mean air temperature of 7.5°C (min. -7 and max. 26.4°C), measured by an automatic weather

station (AWS) mounted above the tree stand during the period from April 2015 to April 2018. The tree cover consisted

predominantly of even-aged Sitka spruce (*Picea sitchensis,* Bong. Carr.) stands. Sitka spruce is the most common conifer in

Great Britain (GB) and represents 26% of the total forested area in GB and 51% of the conifer area (Forestry Commission,

2020. The main soil type is a seasonally waterlogged organic-rich peaty gley classified as histogleysol with a peat (O Horizon)

thickness varying from 15 to 40 cm (Zerva and Mencuccini, 2005), developed in clayey glacial till derived from carboniferous

sediments (Pyatt, 1970).

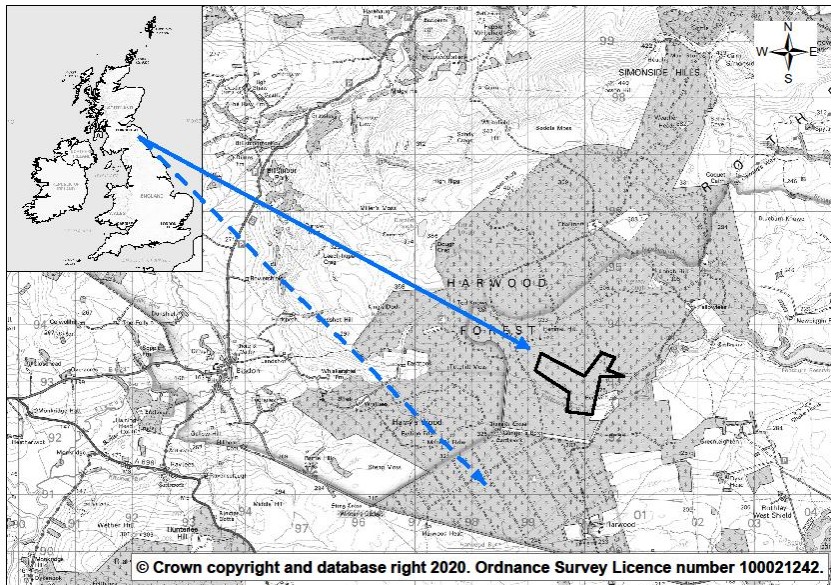

**Figure 1.** Map of experimental areas at Harwood Forest, Northumberland, UK. The dotted arrow shows the unfelled mature

spruce stand area (A), and the solid arrow shows the area that was clearfelled after one year (B), outlined in black.

For the purpose of this study, two nearby areas (A and B) of mature stands within the forest with similar previous

management, soil type and elevation (280 to 290 m) were chosen to carry out measurements over four years between 18

February 2014 to 16 April 2018. During this period, one area (A) was left unfelled (A-yr1 to A-yr4), and the other (B) was

clearfelled after one year (B-yr1 before felling and B-yr2 to B-yr4 after felling). In area A, the 40-ha stand was of second

rotation, even-aged mature Sitka spruce planted in 1973, with yield class of 18 $m^3$ $ha^{-1}$ and mean tree density of 1348 trees $ha^{-1}$. In area B the Sitka spruce stand was planted in 1958, with yield class of 16 $m^3$ $ha^{-1}$ and mean tree density of 1375 trees $ha^{-1}$ prior to felling, and the felled area covered 42 ha. Felling operations were carried out between late January and early March

2015 and followed standard practices of the Forestry Commission (Murgatroyd and Saunders, 2005). Only timber larger than





7 cm diameter was removed from site, leaving tree tops and branches on site in rows, with some used as brash-mats to prevent compaction of soil by the heavy harvesting machinery.

## 2.2 Gas flux measurements and analysis

Forest soil fluxes of $CO_2$, $CH_4$ and $N_2O$ were measured at approximately monthly intervals over the four-year study period from the two areas of the forest. Flux measurements were made using a modified design of the manual static chamber method

described by Yamulki et al. (2013). Each chamber was constructed of opaque PVC with dimensions of 40 cm × 40 cm × 25 cm height to provide a volume of 40 L and placed temporarily on permanently installed frames. The frames were inserted tightly into the ground to a depth of about 3 cm prior to the start of the measurements. The bottom of the chamber had a neoprene rubber foam gasket to ensure a gas-tight seal with the frame, and the top of the chamber had a pressure vent.

Within each area, 8 chambers were positioned randomly in a transect within a 100 $m^2$ area. During each gas flux

measurement, the chambers were placed on top of the frames for up to 60 min and duplicate gas samples of the chamber headspace were taken immediately after closure and then at 3 subsequent 20-min intervals. Gas samples were taken after the chamber was closed by connecting a polypropylene syringe to a chamber sampling port fitted with a three-way stopcock. The syringes were immediately used to fill (under atmospheric pressure) pre-evacuated 20 mL vials fitted with chlorobutyl rubber septa. Concentrations of $CO_2$, $CH_4$ and $N_2O$ were determined within a week using a headspace-sampler (TurboMatrix 110)

and gas chromatograph (GC, Clarus 500, PerkinElmer) equipped with an electron capture detector (ECD) for $N_2O$ analysis, a flame ionization detector (FID) for $CH_4$ analysis, and a catalytic reactor (methanizer) for $CO_2$ analysis by reducing $CO_2$ to $CH_4$ before analysis by the FID detector. The repeatability of the GC gas analysis (assessed as 3 × standard deviation of 20 repeated measurements of standard $CO_2$, $CH_4$ and $N_2O$ concentrations at ambient levels) was better than 4% for all gases.

Gas fluxes were calculated based on linear increases of gas concentrations inside the chambers with time. For $CO_2$

however, if the concentration increase was not linear then fluxes were determined using the R HMR package to plot a best-fit line to the data (Pedersen et al., 2010; R Core Team, 2016) to correct for the non-linearity. We did not apply the HMR model for $CH_4$ and $N_2O$ fluxes as the non-linear fitting plot is very sensitive to variability and outliers in the measured GHG concentrations, particularly for low fluxes and with only four data points per chamber (as also noted by Pihlatie et al., 2013; Brümmer et al., 2017; Korkiakoski et al., 2017) which results in large apparent 'spikes', failure to calculate the fluxes on many

occasions, and likely overestimation of calculated GHG fluxes (Pavelka et al., 2018). If $CO_2$ concentration changes with time were not significant, fluxes for all gases were rejected for that sample as this was judged to be indicative of gas leakage within the chamber headspace.

Although most studies now measure soil $CO_2$ fluxes with IRGAs (as noted by Yashiro et al., 2008), which is viewed as more accurate than gas sampling and GC analysis, we needed to use the GC method to measure all 3 GHG within the

logistical constraints of the experiment. Therefore, to give further assurance in the gas flux calculations, $CO_2$ effluxes were also compared with those from another 25 static chambers (20 cm diameter, 4.2 L volume, LI-8100-103 Survey Chamber, Li-Cor Inc., Lincoln, Nebraska, USA) measured *in situ* in both areas by a closed loop infra-red gas analyser system (IRGA, LI8100A, Li-Cor Inc.), each over a 2-minute duration after chamber closure (Xenakis et al., 2021). These chambers were positioned over a much wider area than the GC chambers to characterise spatial variation and effluxes were measured for three

years but only after felling from Feb 2015. The results (Fig. 2) showed a mean flux difference over the 3-year period of only 6.8% higher by the IRGA method in the unfelled area and 19.5% higher in the felled area compared with the fluxes measured by GC. These are relatively small differences between the two methods, when considering the higher site heterogeneity in the felled area, the inherent differences in the analytical methodologies and as the vegetation in the IRGA chambers was not cut as it was in the GC chambers.




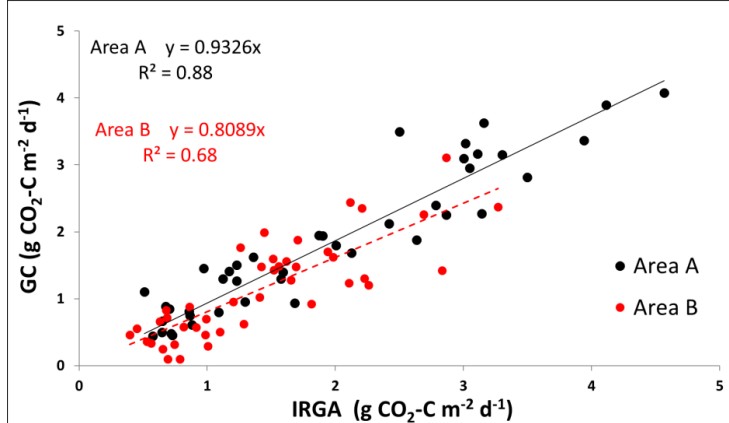

**Figure. 2.** Comparison between mean $CO_2$ effluxes measured by gas chromatograph (GC) from the 8 static chambers in each forest area (A = mature stand, B = clearfell) and mean $CO_2$ effluxes measured by infra-red gas analyser (IRGA), from 25 static chambers in the same areas.


There was no understorey vegetation in the forest stands and no visual evidence of vegetation growth within the chambers in the first two years of the study, with the chamber ground surfaces covered by dense Sitka spruce needle litter. However, in the third year there was growth of Juncus sp. in one chamber in the felled area B between May and September 2016 (maximum height was 40 cm before cutting but the total volume was < 5% of the chamber volume) so the Juncus was

cut continually thereafter. The effect of the vegetation cut on the fluxes was assessed from the flux measurements in all the 8 chambers before and directly after first cutting. The statistical analysis revealed no significant differences between mean chamber fluxes before and after cutting for all gases indicating that variations between the chamber fluxes were greater than that due to cutting. In the third and fourth years there was a proliferation of moss in two of the chambers in area A. This could not be removed without substantial disturbance of the soil surface; comparison of chambers with and without moss showed no

significant differences between mean fluxes. For $CO_2$, the effluxes measured will therefore be from aerobic and anaerobic decomposition processes, and respiration of soil organisms and roots.

**2.3 Soil moisture and temperature measurements**

During each flux sampling day soil temperatures (ºC) at 2 cm and 10 cm depths were recorded from one point around each chamber and volumetric moisture content ($m^3 m^{-3}$) at 6 cm depth was recorded from three points around each chamber. Soil

temperature was recorded by a digital temperature probe (Hanna model Checktemp 1) and the volumetric moisture content by a moisture sensor (SM 200 attached to a handheld HH2 moisture meter, Delta-T Devices Ltd, Cambridge, UK). Sensitivity of $CO_2$ efflux to temperature was determined with a Q10 function (Atkin et al., 2000) which is the proportional change in respiration resulting from a 10°C increase in temperature, derived from data for each year using mean daily values of $CO_2$ efflux. The apparent Q10 values were calculated from the temperature measured near the soil surface (at 2 cm) as recommended

by Pavelka et al. (2007), because the strength of the $CO_2$ efflux and temperature relationship decreases with soil depth.

**2.4 Soil parameters and root biomass measurements**

To assess changes in soil characteristics due to the felling, additional soil parameters that were likely to have an impact on GHG fluxes were measured between 21 and 23$^{rd}$ July 2015, approximately 5 months after the end of felling. Three replicated soil samples were taken from 0-20 cm depth below the litter layer around each chamber at each site for bulk density, pH, total

C and total N content. Soil pH was measured by mixing 5 g soil samples with 25 ml $H_2O$ with analysis using a pH meter probe



(Sentek). Soil C and N stocks were measured by the flash combustion method in an NC Soil Analyser (Flash EA, series 1112, Thermo Scientific). Soil bulk density was measured as the mass of oven-dried soil sample divided by the volume of the cores taken. Soil tree live and dead fine root biomass, length and diameter were measured from 3 replicated samples 0-15 cm depth below the litter layer around each chamber in each site.

**2.5 Statistical analysis**

Statistical analyses were made using statistical package R (R Core Team, 2016). Data were analysed separately for pre-felling and post-felling periods. As measurements were taken from the same eight chambers through time, the analysis was conducted as a repeated measures design, with the significance of felling/non-felling differences determined against the individual chamber data (n=16), not against individual observations. A linear mixed-effects model (Pinheiro et al, 2018) was used to

structure and analyse the data for all flux data. Where necessary, flux data were transformed to obtain a normal distribution. Management type (i.e. felled/unfelled), temperature at 2 and 10 cm and soil moisture (plus all two-way interactions) and date (plus interaction with management type) were treated as fixed effects. Chambers were treated as a random effect (for repeated measures design). For post-felling data, residual analysis indicated that the variance was larger for chambers in the felled area and by individual chamber, therefore weighted variance structures were incorporated within the mixed effects models to

account for within-type and within-chamber heterogeneity.

A range of corARMA (autoregressive moving average) models were applied to each model, to account for temporal autocorrelation within chambers. Analysis of the (partial) autocorrelation function indicated potential temporal autocorrelation up to one previous time point, therefore all combinations of coARMA structure (up to one previous time point) were applied and the best fit model determined using Akaike's Information Criteria (AIC) applied to the maximum likelihood fits across

each of the gas fluxes separately. The significance of the fixed effects were subsequently determined using analysis of deviance (Chi square tests) with non-significant interactions and effects (P>0.05) removed from the final models (Fox and Weisberg, 2011).

Annual cumulative fluxes of $CH_4$, $N_2O$ and $CO_2$ were estimated to assess the inter-annual variations in each area. Within each year, median flux values were calculated across the eight replicate chambers, along with lower and upper quartiles,

maximum and minimum values. These values were then accumulated across the year, by taking the mean of two consecutive flux values and multiplying it by the number of days between the measurements and summing over the monitoring period for each 12 months as indicated in Table 1 and adjusting to 365 days. For year one where the felling interrupted the measurement in area B, the same cumulative time period was taken for both areas. Annual cumulative fluxes were converted into $CO_2$ equivalent ($CO_2e$) using the global warning potential (GWP) for a 100-year time horizon of 34 for $CH_4$ and 298 for $N_2O$ (IPCC,

210 2013).

**Table 1.** Measurement periods used for the statistical analysis and calculations of the cumulative soil fluxes at Harwood Forest, Northumberland, UK, before adjusting to 365-day calendar year.

| Year | Measurement period | | Period length |
|---|---|---|---|
| | Start | End | |
| 1 Pre-felling | 18-02-2014 | 27-01-2015 | 343 |
| 2 | 21-04-2015 | 06-04-2016 | 351 |
| 3 | 06-04-2016 | 05-04-2017 | 364 |
| 4 | 05-04-2017 | 16-04-2018 | 376 |


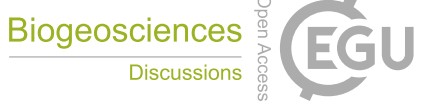

## 3 Results

### 3.1 Soil temperature, moisture

Over the 4 years, soil temperature measured during gas sampling varied between 0.1 and 26.0 ºC (mean 9.7 ºC) at 2 cm soil depth (Fig. 3a) and 1.3º and 15.5 ºC (mean 8.3 ºC) at 10 cm depth (data not shown), although soil temperatures were never above 15ºC under the trees in area A. Before felling in year 1, there were only small differences (p < 0.001) in mean soil temperature between areas A and B (7.1º and 8.5 ºC,  respectively) at 2 cm soil depth and at 10 cm (7.1º and 7.9 ºC)  (Table 2), probably caused by sampling the area B later in the day than A. After felling, the mean soil temperature increased in the felled area at 2 cm soil depth (mean 14.3º compared to 9.2 ºC in the unfelled area, p<0.001), and at 10 cm depth (10.2 ºC compared to 8.4 ºC in unfelled), due to removal of the tree cover (Xenakis et al., 2021). In the following two years, the soil temperature remained higher in the felled area by up to 3.4 ºC compared to area A, at the 2 cm depth (1.6 ºC at 10 cm) and was on average 2 ºC higher than before felling (there was no change in the mean soil temperature in A).

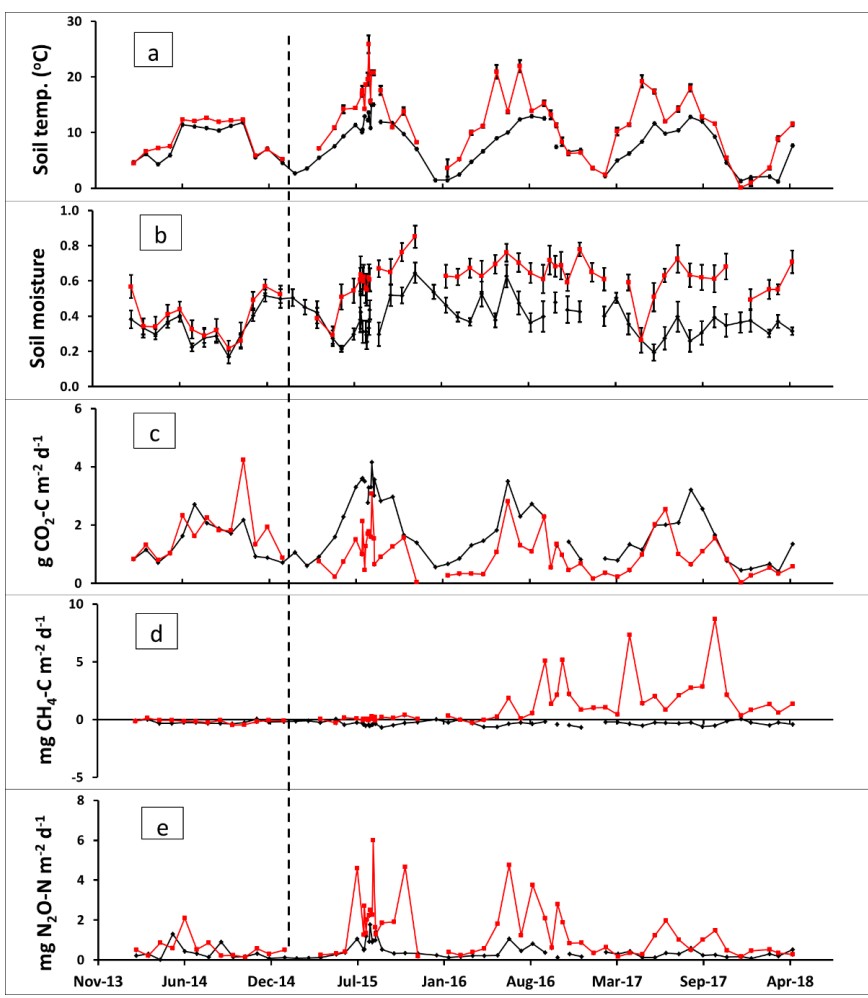

**Figure 3**. Mean soil temperature at 2 cm depth (a), volumetric soil moisture content (cm$^3$ cm$^{-3}$) (b), and median soil CO$_2$ (c), CH$_4$ (d), and N$_2$O (e) fluxes measured approximately monthly throughout the experiment in Harwood Forest. Black symbols and lines are for area A,  (mature spruce stand) and red are for area B, clearfelled after one year as indicated by the dotted vertical line. Error bars are standard error of mean of 8 replicate measurements of the soil temperatures and moisture.





**Table 2.** Mean of 8 replicate measurements (±SE) of soil temperatures at 2 cm and 10 cm depths and moisture at 6 cm depth
measured close to the soil GHG flux chambers in area A (mature spruce stand) and area B (clearfell area after year 1) in
Harwood Forest.

| Year | Soil Temp. 2cm ºC | | Soil Temp. 10cm ºC | | Soil moist. $m^3 m^{-3}$ | |
|---|---|---|---|---|---|---|
| | A | B | A | B | A | B |
| 1 | 7.09 (0.10) | 8.52 (0.14) | 7.12 (0.1) | 7.90 (0.13) | 0.36 (0.04) | 0.39 (0.05) |
| 2 | 9.22 (0.08) | 14.29 (0.52) | 8.40 (0.08) | 10.21 (0.16) | 0.38 (0.05) | 0.60 (0.07) |
| 3 | 7.89 (0.08) | 11.25 (0.46) | 7.55 (0.06) | 8.78 (0.19) | 0.44 (0.06) | 0.67 (0.06) |
| 4 | 7.09 (0.09) | 10.52 (0.37) | 6.88 (0.09) | 8.80 (0.21) | 0.32 (0.06) | 0.58 (0.06) |

No significant differences in soil moisture content (by volume) were observed between the two areas before felling
with a mean of 0.36 in area A and 0.39 $m^3 m^{-3}$ in B in year 1 (Fig. 3b and Table 2). However, after felling the soil moisture
content was significantly higher in the felled area than the unfelled (mean 0.62 $m^3 m^{-3}$ cf. 0.38 $m^3 m^{-3}$, p<0.001), due to the
reduced evapotranspiration after tree removal (Xenakis et al., 2021). In both areas, there was a pronounced seasonal variation
in soil moisture in the year before felling (2014) and in the first few months of 2015, with higher moisture in winter months
than in summer, but this pattern was not clear thereafter because the rainfall was more evenly distributed during the last two
years of this study (Fig. 4).


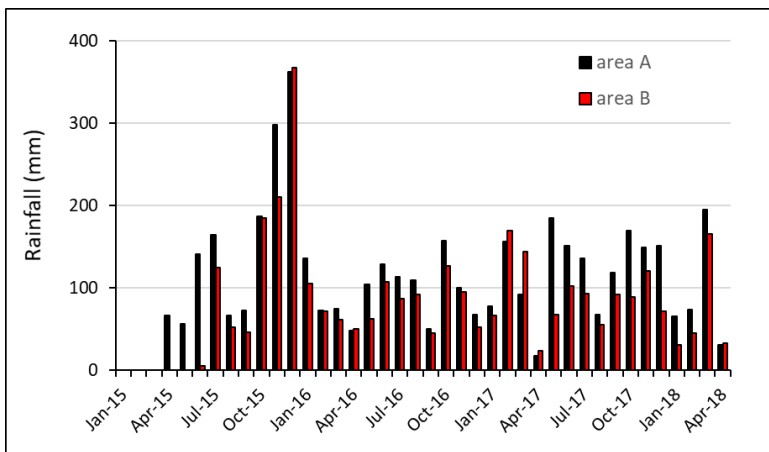

**Figure 4.** Monthly total rainfall measured after felling in each area in Harwood Forest, atop the 30 m tower in the mature
spruce stand (area A, black), and on the ground in the clearfell site (area B, red). Measurements only started in area B on
June 2015.

**3.2 Soil parameters and root biomass measurements**

Soil parameters for the 0-15 cm layer measured 5 months after felling in year 2 showed no significant differences between the
unfelled and felled areas in mean soil pH (3.6 and 3.8, respectively) and soil total N content (1.8 and 1.95 %) but the mean soil
total C content was about 17% lower (p <0.001) in the felled area (52.0 and 43.3 %). The soil bulk density was significantly
(p <0.001) higher at 0.30 g $cm^{-3}$ in the felled area compared with 0.22 g $cm^{-3}$ in the unfelled, but both values are typical for
peaty gley soils (Vanguelova et al., 2013). Mean fine (< 2mm) live root mass was much lower in the felled area, as expected
(1.6 t $ha^{-1}$ cf. 4.9 t $ha^{-1}$), but the difference was smaller for the mean fine dead root mass (0.35 and 0.53 t $ha^{-1}$, respectively),
probably due to partial or complete decomposition during the 5-months after felling prior to the measurements.





### 3.3 GHG fluxes

Large variations were observed in the fluxes between the 8 replicate chambers after felling with some high outliers, particularly for $CH_4$. Therefore, to reduce bias in the annual budget estimates of $CO_2$, $CH_4$ and $N_2O$ fluxes and enable a robust comparison between annual fluxes before and after felling, the annual and cumulative fluxes were based on the median of the replicate chambers as described in the statistical analysis section

### 3.3.1 $CO_2$ effluxes

In the first year before felling, there were no significant differences in soil $CO_2$ effluxes between areas A and B (median 1.54
and 1.75 g $CO_2$-C m$^{-2}$ d$^{-1}$, respectively, Fig. 3c). In the following 3 years after felling, $CO_2$ effluxes became significantly (p<0.001, Table 3) lower in the felled area (median 1.10, 0.90, and 0.92 g $CO_2$-C m$^{-2}$ d$^{-1}$ in year 2, 3 and 4 respectively) than in the unfelled area (2.44, 1.69 and 1.44 g $CO_2$-C m$^{-2}$ d$^{-1}$). There was a clear seasonal variation in the $CO_2$ effluxes at both areas, which as expected followed that of soil temperature with maximum effluxes during June to September.

**Table 3.** Results from the analysis of deviance (Chi square tests) showing the probability (p) values for the effects of the explanatory factors/variables: felling, soil temperature (at two depths 2 and 10 cm), volumetric soil moisture measurement date and their interactions (as fixed effects) on soil $CO_2$, $CH_4$ and $N_2O$ fluxes in Harwood Forest. Note: p < 0.05 is deemed significant, Pre-fell denotes comparison between area A (mature spruce stand) and area B (clearfell) before felling in year 1, Post felling denotes comparisons between the areas for year 2 to 4.

| Gas | variables | Pre-felling | Post-felling |
|---|---|---|---|
| $CO_2$ | Area | 0.352 | <0.001 |
| | Temp.2 | 1.000 | 0.587 |
| | Temp.10 | <0.001 | <0.001 |
| | Moisture | 0.637 | 0.850 |
| | Area:Temp.2 | 0.140 | <0.001 |
| | Area:Temp.10 | 0.200 | 0.232 |
| | Area:Moisture | 0.986 | 0.600 |
| | Temp.2:Temp.10 | 0.083 | 0.170 |
| | Temp.2:Moisture | 0.474 | 0.641 |
| | Temp.10:Moisture | 0.346 | 0.847 |
| | | | |
| $CH_4$ | Area | 0.803 | <0.001 |
| | Temp.2 | 0.542 | 0.002 |
| | Temp.10 | 0.436 | 0.024 |
| | Moisture | 0.696 | 0.086 |
| | Area:Temp.2 | 0.959 | 0.574 |
| | Area:Temp.10 | 0.953 | 0.547 |
| | Area:Moisture | 0.205 | 0.149 |
| | Temp.2:Temp.10 | 0.392 | 0.170 |
| | Temp.2:Moisture | 0.882 | 0.341 |
| | Temp.10:Moisture | 0.766 | 0.605 |
| | | | |
| $N_2O$ | Area | 0.588 | 0.010 |
| | Temp.2 | 0.269 | 0.091 |
| | Temp.10 | <0.00 | <0.001 |
| | Moisture | 0.002 | 0.407 |
| | Area:Temp.2 | 0.034 | 0.134 |
| | Area:Temp.10 | 0.062 | 0.024 |
| | Area:Moisture | 0.031 | 0.427 |
| | Temp.2:Temp.10 | 0.967 | 0.100 |
| | Temp.2:Moisture | 0.195 | 0.192 |
| | Temp.10:Moisture | 0.007 | 0.081 |


There was no significant correlation between $CO_2$ effluxes and soil moisture, in part because in the felled area the moisture was high most of the time (Fig. 3b), although low $CO_2$ effluxes were generally associated with high soil moisture

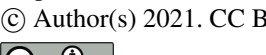



(and lower temperatures) in the winter and the highest effluxes were observed during low soil moisture periods in the warmer

temperatures of the summer, particularly in the unfelled area. $CO_2$ effluxes increased ($p \leq 0.002$) with soil temperature at the

2 and 10 cm depths at both areas which was best described by exponential correlation relationships as shown in Fig. 5 for the

periods before and after felling. Before felling, the $CO_2$ efflux response to soil temperature was not significantly different

between area A and B, with comparable Q10 values (3.77 and 3.16, respectively, Table 4). However, in the years after felling,

the Q10 values became much lower in the felled area B (mean Q10 = 2.7) than in the unfelled area A (mean Q10= 4.23) with

the lowest Q10 value of 2.1 in year 3.

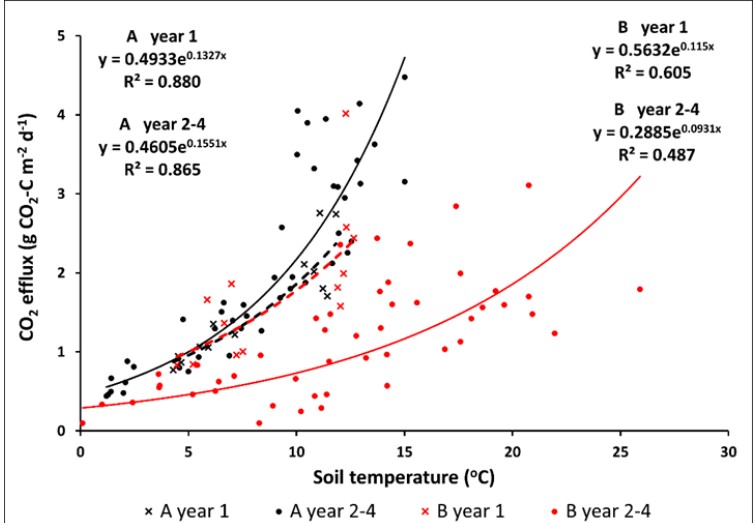

**Figure 5.** Exponential relationship between soil $CO_2$ effluxes and soil temperature at 2 cm depth measured from area A

(mature spruce stand, black) and area B (clearfell site, red), before (dashed lines, 'yr1') and after (solid lines, 'yr2-4') felling.


**Table 4.** Annual apparent Q10 values for soil $CO_2$ effluxes in area A (mature spruce stand) and area B (clearfell area after year 1) in Harwood

Forest during the study period (as defined in Table 1).

| Year | A (unfelled) | | | B (felled after yr1) | | |
|---|---|---|---|---|---|---|
| | $R^2$ | slope | Q10 | $R^2$ | slope | Q10 |
| 1 | 0.880 | 0.133 | 3.77 | 0.605 | 0.115 | 3.16 |
| 2 | 0.866 | 0.149 | 4.44 | 0.486 | 0.094 | 2.57 |
| 3 | 0.700 | 0.120 | 3.30 | 0.319 | 0.072 | 2.06 |
| 4 | 0.973 | 0.160 | 4.94 | 0.654 | 0.124 | 3.46 |


The annual cumulative $CO_2$ effluxes from the felled and unfelled areas were not significantly different before felling

with large overlap between the 95% confidence intervals (Fig. 6a), and the annual $CO_2$ effluxes were 19.8 and 24.0 t $CO_2$e ha$^{-1}$ yr$^{-1}$ in areas A and B, respectively (Table 5). After felling, however, there was a clear divergence in the effluxes between the

areas, with the $CO_2$ efflux dropping sharply in the felled area (B). In the first year after felling, the annual $CO_2$ efflux reduced

to its minimum value of 8.9 compared with 23.0 t $CO_2$e ha$^{-1}$ yr$^{-1}$ from the unfelled area A. In years 3 and 4 the annual effluxes

in the felled area increased gradually to 11.3 and 12.2 t $CO_2$e ha$^{-1}$ yr$^{-1}$, respectively, but was still lower than the unfelled area

(20.3 and 18.4 t $CO_2$e ha$^{-1}$ yr$^{-1}$).





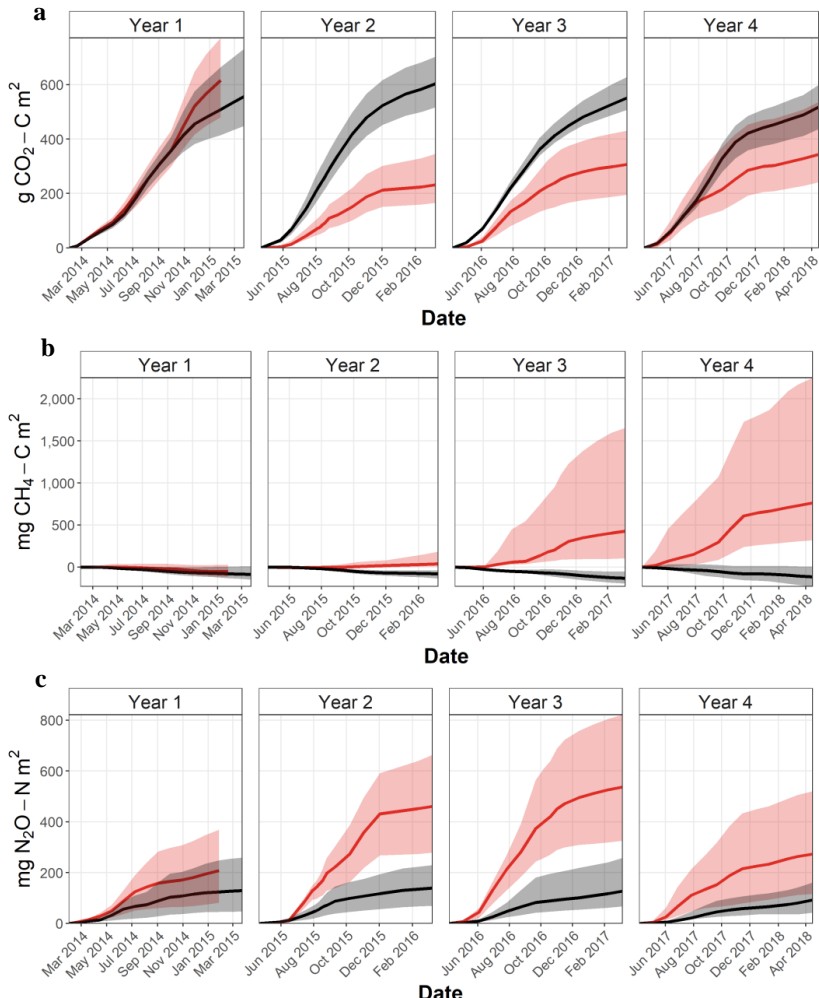

**Figure 6.** Cumulative soil fluxes of $CO_2$ (a), $CH_4$ (b) and $N_2O$ (c) in Harwood Forest from area A (black, mature spruce stand) and area B (red, clearfelled after first year) during each year, calculated from median fluxes of 8 replicated chambers. Ribbons are the estimated 95% confidence intervals.

**Table 5.** Annual soil GHG fluxes expressed as Global Warming Potential (GWP, t $CO_2$e ha-1 $yr^{-1}$) in area A (mature spruce stand) and area B (clearfell area after year 1) in Harwood Forest. Values in brackets are the % contribution to total GWP.

| Year | $CO_2$ | | $CH_4$ | | $N_2O$ | | Total | |
|---|---|---|---|---|---|---|---|---|
| | A | B | A | B | A | B | A | B |
| 1 | 19.82 (97.1) | 24.03 (96.0) | -0.038 (-0.2) | -0.026 (-0.1) | 0.620 (3.0) | 1.034 (4.1) | 20.41 | 25.04 |
| 2 | 23.03 (97.3) | 8.87 (79.7) | -0.038 (-0.2) | 0.018 (0.2) | 0.678 (2.9) | 2.246 (20.2) | 23.67 | 11.13 |
| 3 | 20.29 (97.4) | 11.30 (80.6) | -0.060 (-0.3) | 0.194 (1.4) | 0.596 (2.9) | 2.524 (18.0) | 20.83 | 14.01 |
| 4 | 18.44 (98.0) | 12.24 (88.6) | -0.051 (-0.3) | 0.335 (2.4) | 0.419 (2.2) | 1.243 (9.0) | 18.80 | 13.82 |



### 3.3.2 CH$_4$ fluxes

Soil fluxes of CH$_4$ in both areas of the forest were generally low throughout the study period, particularly before felling with no significant differences (Fig. 3d). Fluxes were predominantly negative (i.e. removal from the atmosphere) in unfelled area A and before felling in area B with a median flux of -0.33 mg CH$_4$-C m$^{-2}$ d$^{-1}$ and -0.21 mg CH$_4$-C m$^{-2}$ d$^{-1}$, respectively. After felling, area B became a significant (p <0.001) source of CH$_4$ and fluxes increased rapidly in the following 2 years to its maximum in year 4 (2.48 mg CH$_4$-C m$^{-2}$ d$^{-1}$) compared to the unfelled area, which remained unchanged with a small CH$_4$ sink
(-0.33 mg CH$_4$-C m$^{-2}$ d$^{-1}$).

Fluxes of CH$_4$ varied more between the flux chambers after felling particularly in year 3 and 4 (Fig. 6b). Although both soil moisture and CH$_4$ fluxes increased in the felled area after felling, the increased fluxes and the variation between chambers cannot directly be related to the soil moisture as no significant overall correlation was observed. In addition, including interactions between soil moisture and temperature in the statistical model did not better explain the variation in CH$_4$
fluxes and difference between areas (Table 3). However, the analysis showed that CH$_4$ fluxes after felling were best explained by the soil temperature (negative association) at both the 2 cm and 10 cm depths.

Annual fluxes (as GWP) of CH$_4$ in the first period before felling were -0.038 and -0.026 t CO$_2$e ha$^{-1}$ yr$^{-1}$ from areas A and B, respectively (Table 5). In the following years, the felled area (B) became a consistent source of CH$_4$ of 0.018, 0.194, and 0.335 t CO$_2$e ha$^{-1}$ yr$^{-1}$ in year 2, 3 and 4 respectively. In contrast, the unfelled area (A) remained a small CH$_4$ sink with a
mean GWP value of -0.050 t CO$_2$e ha$^{-1}$ yr$^{-1}$.

### 3.3.3 N$_2$O fluxes

There were no significant differences in N$_2$O fluxes between the two areas before felling (Fig. 3e), although the median flux was higher in B (0.33 mg N$_2$O-N m$^{-2}$ d$^{-1}$ and  0.55 mg N$_2$O-N m$^{-2}$ d$^{-1}$ for A and B respectively). Maximum N$_2$O fluxes of 1.30 and 2.12 mg N$_2$O-N m$^{-2}$ d$^{-1}$ were measured from area A and B, respectively, between May and June 2014. After felling, N$_2$O
fluxes in the felled area B became significantly (p = 0.01) higher in year 2, 3, and 4 (median 1.83, 1.45, and 0.72 mg N$_2$O-N m$^{-2}$ d$^{-1}$) than in the unfelled area A (0.63, 0.39, and 0.28 mg N$_2$O-N m$^{-2}$ d$^{-1}$) with a maximum flux of 6.01 mg N$_2$O-N m$^{-2}$ d$^{-1}$ measured in the first year after felling in August 2015.

There were no significant correlations between N$_2$O fluxes and soil temperature before felling. However, after felling, N$_2$O fluxes showed a seasonal pattern that followed (p <0.001) that of soil temperature at both depths in both areas with
maximum fluxes during periods from June to October. Soil moisture and its interactions with soil temperature (at 10 cm) were the main driver for N$_2$O fluxes before felling (p = 0.002 and p = 0.007, respectively). After felling, however, the soil moisture remained high (Fig. 3b) and no direct effect on N$_2$O fluxes was observed. This could be due to the significant (p < 0.01) negative correlation between soil temperature (at both depths) and moisture before felling, compared with after felling where the seasonal pattern in soil moisture was less clear (Fig. 3b and Fig. 4) and no significant correlation occurred, so that the soil
temperature became the main driver of N$_2$O fluxes.

There was a large variation between chambers in N$_2$O fluxes throughout the study period in both areas of the forest (evident in the confidence intervals shown in Fig. 6c). Before felling, N$_2$O annual fluxes (expressed as GWP) were 0.62 and 1.03 t CO$_2$e ha$^{-1}$ yr$^{-1}$ in areas A and B, respectively (Table 5), much smaller than the GWP contribution from CO$_2$ effluxes. After felling, the annual fluxes of N$_2$O increased and the highest annual fluxes were measured from the felled area in the two
consecutive years after felling with 2.25 and 2.52 t CO$_2$e ha$^{-1}$ yr$^{-1}$ in year 2 and 3 respectively. However, at the end of the monitoring period in year 4, the annual flux of N$_2$O returned to a rate similar to that before felling, (1.24 t CO$_2$e ha$^{-1}$ yr$^{-1}$).





## 4 Discussion

### 4.1 Effect of felling on CO₂ effluxes

Felling and removal of the trees reduced soil $CO_2$ effluxes by 55%, comparing the mean over the 3 years post-felling with that
pre-felling, or 47%, comparing the clearfelled with the mature stand (Table 5). Presumably this was a consequence of the
reduction in autotrophic root and rhizosphere respiration (e.g. Boone et al., 1998, Takakai et al., 2008), and measurements
about 5 months after felling showed a reduction from 4.9 t ha$^{-1}$ to 1.6 t ha$^{-1}$ in the live fine root mass (i.e. diameter <2mm).
Living fine roots and their associated mycorrhizae can contribute up to 59% of total respiration (Ewel et al., 1987; Irvine and
Law, 2002; Subke et al., 2006), and for similar spruce stands in Harwood Forest about 40% contribution has been estimated
previously (Zerva and Mencuccini, 2005). However, $CO_2$ effluxes might also increase directly after felling due to an increase
in decomposition of fine roots and associated ectomycorrhizal biomass and litter. Therefore, the net $CO_2$ efflux will be
determined by the balance between the reduction in autotrophic root and rhizosphere respiration and the increased
decomposition (Köster et al., 2013) which can be short-lived depending on environmental factors such as soil temperature and
moisture (Davidson et al., 1998; Skopp et al., 1990).

Before felling and in the unfelled stand, $CO_2$ effluxes showed a strong seasonal pattern that followed that of soil
temperature with higher effluxes during summer, as expected, when fine root density and plant growth activity are highest.
The apparent Q10 values of these $CO_2$ effluxes (Table 4) were lower than the value reported from a larch forest in eastern
Siberia (5.92; Takakai et al., 2008), but at the higher end of those reported from a temperate deciduous oak forest in UK (1.60
to 3.92; Yamulki and Morison, 2017). The response of $CO_2$ efflux to temperature became weaker after felling (Fig. 5), even
though the soil temperature was substantially higher (a decrease in the apparent Q10 values by up to 36% over the 3-year
period; Table 4). This agrees with the study of Zerva and Mencuccini, (2005) who also observed a weaker association of $CO_2$
effluxes with soil temperature over a 10 month period after felling at another site in this forest. They noted that this was
probably because of the increased water content (also evident in our study, Fig. 3b), the death of fine roots and the disturbance
of the soil caused by tree harvesting, and suggested that autotrophic root respiration was more responsive to temperature than
heterotrophic microbial respiration. However, this effect could also be because the apparent Q10 of soil $CO_2$ efflux determined
from field measurements like these with trees present is influenced by the seasonal changes in radiation and photosynthesis
and their positive association with seasonal temperature, rather than an altered temperature sensitivity after felling. A larger
reduction of 64% in the Q10 value of soil $CO_2$ efflux, compared to that found in this study, was reported from a larch forest in
eastern Siberia comparing a disturbed clear-cut site with a forest site (2.1 cf. 5.9, Takakai et al., 2008). At the end of our study
period in year 4, the Q10 value for soil $CO_2$ efflux in the felled area (3.46) was close to that of before felling (3.16 in B-yr1),
which could be due to ground vegetation growth near but outside the gas flux chambers. It may also be a result of a drop in
the rainfall and soil moisture in the period between May and June in year 4 (Fig.3b, Fig. 4) so the response to soil temperature
became stronger, as in the period before felling. There may also have been a recovery from any compaction effects during
harvesting, as Epron et al. (2016) showed that compaction by timber forwarding machinery after harvesting a French oak forest
on a mineral soil decreased the Q10 values of soil $CO_2$ by 16-22%. In contrast, Kulmala et al. (2014) found that the temperature
dependency of the $CO_2$ efflux was not affected by clear-cutting of a Norway spruce forest on organo-mineral soil in southern
Finland.

The response of $CO_2$ efflux to soil temperature is likely to have been affected by the substantial increase in soil
moisture after felling (Fig 3b) as noted by Zerva and Mencuccini (2005) and Kulmala et al. (2014). This increase in soil
moisture (with values frequently > 0.6 cm$^3$ cm$^{-3}$) could have affected soil respiration by limiting the diffusion of substrates and
$O_2$ to microorganisms (Skopp et al., 1990). Yamulki and Morison (2017) could not detect an effect of soil moisture alone on
soil respiration in an oak forest in SE England, but the combined model of soil temperature and moisture explained 73% of the
$CO_2$ efflux variations. It is also possible that some of the $CO_2$ produced may have dissolved in the soil water and gone



undetected (Zerva and Mencuccini, 2005), but probably this effect was negligible here because of the low solubility of $CO_2$ at
the low soil pH (3.8).

### 4.2 Effect of felling on $CH_4$ fluxes

Clearfelling changed the soil from a small annual sink of $CH_4$ to a small net source over the 3-year monitoring period after
felling, while the unfelled stand remained a sink in all years. The shift in $CH_4$ fluxes from net uptake to net emissions by
clearfelling has been reported in several other studies: Zerva and Mencuccini (2005) from another site within this spruce forest;
Castro et al. (2000) from two slash pine plantations in Florida; Takakai et al. (2008) from a Siberian larch forest soil; Yashiro
et al. (2008) from a tropical rain forest in Malaysia; Sundqvist et al. (2014) from a forest site in central Sweden; and more
recently by Korkiakoski (2019) from Scots pine nutrient-rich peatland forest in southern Finland. As soil $CH_4$ production
requires anaerobic conditions (Conrad, 2007), this shift from sink to source was probably caused by the substantial increase in
soil moisture (62% higher) and soil temperature (6 ºC) in the first year after felling. An increase in soil moisture can increase
the anaerobic conditions that favour $CH_4$ production by methanogenic archea (e.g. Sundqvist et al., 2014) and therefore can
change the direction of the $CH_4$ flux. Generally, soil temperature and particularly moisture are considered to be good predictors
for $CH_4$ behaviour (Lavoie et al., 2013) but disentangling the two influences is difficult in field conditions.

In this study, $CH_4$ fluxes and the flux variations between chambers increased significantly in the following years after
felling. The increase in fluxes was modest in the first year after felling but more substantial in the second and third years (Fig.
6b) which may reflect a time lag in the microbial community changing. Although the increase in fluxes might be attributed to
the substantial increase in the soil moisture, there was no direct correlation between $CH_4$ fluxes and soil moisture. Some
previous studies have also attributed the increase in $CH_4$ fluxes after felling to an increase in soil moisture or rise in the water
table (Sundqvist et al., 2014; Zerva and Mencuccini, 2005; Korkiakoski et al., 2019; Epron et al., 2016) while some observed
no direct correlations between $CH_4$ fluxes and soil moisture after felling (Lavoie et al., 2013; Takakai et al., 2008; Wu et al.,
2011; Mäkiranta et al., 2012; Sundqvist et al., 2014 and Zerva and Mencuccini, 2005). The lack of direct correlation between
$CH_4$ fluxes and soil moisture has previously been attributed to: i) insensitivity of methanotrophic activity to small variations
in soil moisture and temperature (Sjögersten and Wookey, 2002; Peichl et al., 2010; Wu et al., 2011; Mäkiranta et al., 2012)
particularly when fluxes are relatively low such as in this study; ii) $CH_4$ being produced at a depth greater than that of the
measured soil moisture (Zerva and Mencuccini, 2005), and iii) to other overriding biological and physical factors (Lavoie et
al., 2013).

Soil temperature determines $CH_4$ fluxes by influencing methanogenic and methanotrophic activity differently (Luo
et al., 2013; Aronson et al., 2013). Generally, $CH_4$ consumption by methanotrophs is less responsive to temperature than $CH_4$
production by methanogens as consumption is mainly limited by atmospheric $CH_4$ diffusion (Dunfield et al., 1993; Kruse et
al., 1996). This is in line with the results here, as the statistical analysis showed soil temperature better explained $CH_4$ fluxes
after felling when it became a source than before when the soil was a $CH_4$ sink. However, soil temperature can be positively
related to $CH_4$ uptake (Maljanen et al., 2003; Wu et al., 2011; Ullah and Moore, 2011; Yamulki and Morison, 2017), emissions
(Zerva and Mencuccini, 2005; Dunfield et al., 1993; Ullah and Moore, 2011), or can have no correlation (Takakai et al., 2008,
Sjögersten and Wookey, 2009; Lavoie et al., 2013) depending on soil moisture and other factors that affect microbial $CH_4$
production and consumption. We found no significant interactive effect between temperature and moisture on $CH_4$ fluxes here.

Clearfelling can also affect other factors that play a key role in microbial $CH_4$ production or oxidation (and $CO_2$ and $N_2O$
fluxes) by increasing the substrate availability and soil N ($NH_{4+}$ and $NO_3^-$) concentrations (e.g. Bradford et al., 2000; Wang
and Ineson, 2003), for example, as a result of nitrogen release from litter or brash; or reduced soil pH (Dalal and Allen, 2008).
Felling increased the bulk density from 0.22 to 0.30 g m$^{-3}$ by compaction as a result of the machinery traffic which can
contribute to reduced $CH_4$ uptake, increased $CH_4$ production and/or $CH_4$ release (Teepe et al., 2004; Frey et al., 2011).
However, the change in soil substrates (organic matter and microorganisms) after felling was not measured in this study, and





the differences between the unfelled and felled area for total soil N content (1.95% and 1.83%, respectively) and soil pH (3.6 and 3.8 respectively) were small.

### 4.3 Effect of felling on N₂O fluxes

N₂O fluxes in temperate forest soil are generally expected to be low because of the high C:N ratios in the litter and topsoil (Butterbach-Bahl and Kiese, 2005; Jarvis et al., 2009) but can have a high spatial variability because of the variability in the controlling environmental factors (Peichl et al., 2010; Fest et al., 2009). In this study, both areas of the forest showed large between-chamber variation in N₂O fluxes throughout the study period. This could be due to variations in soil moisture *within* different flux chambers, particularly after rainfall, and the variability between chambers in soil characteristics, litter amount, mineral N availability and microbial biomass, which were not measured. N₂O fluxes (Fig. 3e) increased after felling

and there was an association with soil temperature at 10 cm depth (Table 3). Soil moisture was a significant driver for N₂O fluxes before felling, but it became less significant after felling. As the soil moisture was consistently higher after felling (Table 2), the lack of relationship with soil N₂O fluxes may indicate that the moisture content was not limiting for soil N₂O production by the main microbial nitrification and denitrification processes and that other factors were responsible for N₂O flux variations. As mentioned previously, the total soil N content measured some 5 months after felling was very similar to that in the unfelled

area. However, microbial N₂O production in the following years after felling was probably influenced by a declining slow N release into the soil from the decomposition of fresh tree harvest residues and roots (Zerva and Mencuccini, 2005; Yashiro et al., 2008; Saari et al., 2009) but stimulated by warmer soil temperatures in the summer (Kulmala et al., 2014). This is consistent with the significant relationship between N₂O fluxes and soil temperature, particularly at the 10 cm depth, which might imply that N₂O production was more prominent at the deeper, more anaerobic soil depth, probably caused by denitrification brought

about by an increased respiratory sink for $O_2$. In the absence of plants, there is no competition for this newly available N, thereby maximizing substrate availability for microbial N₂O production and release (Skiba et al., 2012). Three-fold higher N₂O emissions from Finnish drained peatland pine forest plots with logging residues than from unlogged have been reported (Mäkiranta et al., 2012). Such effects of increasing soil temperature combined with microbial activities and microbial biomass N in increasing N₂O fluxes have also been reported by other studies (Ineson, et al., 1991; Smolander et al.,1998; Zerva and

Mencuccini, 2005; Papen and Butterbach-Bahl, 1999; Smith et al., 2018).

The effect of clearfelling in increasing N₂O fluxes by similar magnitudes has also been shown from forests on both mineral and peat soils (e.g. Saari et al., 2009; Zerva and Mencuccini, 2005; Mäkiranta et al., 2012; Pearson et al., 2012), but an order of magnitude higher N₂O flux was measured after clearfelling a nutrient-rich drained peatland forest in southern Finland (Korkiakoski et al., 2019). It is pertinent to mention here the study of Liimatainen et al. (2018) from a range of

afforested northern peat soils in Finland, Sweden and Iceland, where they suggested that high N₂O fluxes were linked to availability of peat phosphorus (P) and copper (Cu) which could be released with other nutrients by harvesting from soil disturbance and brash (Rodgers et al., 2010), and that low P and Cu concentrations can limit N₂O production even with sufficient N availability.

### 4.4 Clearfell harvesting effect on the GHG balance

Currently, emissions resulting from forest management, such as felling and thinning are not considered by IPCC Tier 1 and 2 LULUCF inventories, although these management activities could potentially change emission rates. In this study, the effect of clearfelling on the overall soil GHG budget and the contributions of each gas to the total GHG soil flux were assessed by comparing the annual fluxes using their global warming potential (GWP, Table 5). Before felling, the total GWP (sum of $CO_2$, $CH_4$ and N₂O in t $CO_2$e ha$^{-1}$) were 20.4 in A and 25.0 in B areas and $CO_2$ effluxes dominated the total GHG GWP, contributing

up to 97%. The GWP dropped in the first and second year after felling to approximately half (44% and 56% lower annual effluxes, respectively). The contribution of $CO_2$ to the total flux in the unfelled area remained constant at about 98% throughout

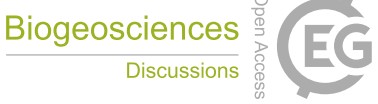

the study period but decreased in the felled area to ca. 80% in the first and second year after felling. This was due to the doubling of $N_2O$ flux which contributed up to 20% of the total GWP in the two years after felling. Although the felled site became a source of $CH_4$, its contribution to the GWP was always small (<2%). For the same periods, the contributions of $N_2O$

and $CH_4$ to the total GWP in the unfelled area were small and remained constant at 2.9% and -0.2% respectively, similar to their values in year 1. In the last year, $N_2O$ annual efflux in the felled area halved to 1.2 t $CO_2e$ ha$^{-1}$, still 20% higher than before felling, but efflux of $CH_4$ continued to increase to 0.34 t $CO_2e$ ha$^{-1}$. Over the 3 years since felling, the total soil GHG flux GWP was reduced by 45% (from 25.0 to 13.8 t $CO_2e$ ha$^{-1}$) due to the much larger reduction in soil $CO_2$ efflux than the increases in $N_2O$ and $CH_4$ fluxes. In the unfelled area there was a reduction of approximately 8%, presumably due to changing

weather conditions. Fig. 7. summarises the overall results by showing the ratio of the annual soil GHG fluxes in B to that in A, with the assumption that the year-to-year variation in the unfelled area A is an indicator equally of what conditions would have been for B in the absence of felling. The figure shows a clear annual GHG flux response to felling where $CO_2$ effluxes reduced directly after felling, increasing gradually thereafter; $CH_4$ effluxes increased sharply in year 3 and 4 after felling; and $N_2O$ effluxes increased in year 2 and 3 after felling but decreased thereafter.


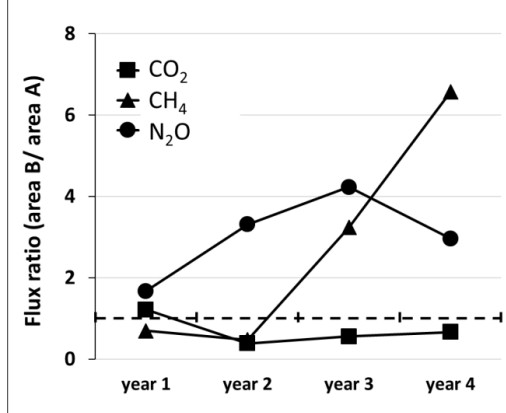

**Figure 7.** Ratios of annual soil GHG fluxes in area B (clearfell area, felled after year 1) to that in area A (mature spruce stand) in Harwood Forest. Note: $CH_4$ fluxes presented as absolute values. Dashed line is for ratio = 1 (no difference).

This study is one of very few longer-term assessments of the impacts of clearfell harvesting on the GHG balance of Sitka spruce forest. However, because of the limitations of the periodic manual closed chamber measurement technique used here, it does not take into account the daily temporal flux variations or cover fluxes from ground occupied by brash or stumps. Simultaneous eddy covariance (EC) measurements (Xenakis et al., 2021) over the 3-year period after clearfelling at this study area showed approximately 3 times higher ecosystem respiration (32.8 t $CO_2$ ha$^{-1}$ yr$^{-1}$) than our estimated mean annual soil

$CO_2$ efflux (10.8 t $CO_2$ ha$^{-1}$ yr$^{-1}$). The difference presumably indicates high $CO_2$ effluxes coming from the brash mats and stumps, as also suggested by Zerva and Mencuccini (2005), which could not be measured by the small chambers used in this study plus above and belowground respiration of the colonising vegetation which was not included in the chambers. The differences could also be due to spatial heterogeneity over the site as the soil flux chambers are only partially sampling ground that is representative of the EC footprint. Moreover, it has been indicated that peat decomposition after felling is stimulated by

nutrient release from brash (Vanguelova et al., 2010) and therefore higher $N_2O$ and $CH_4$ fluxes may also be expected as a result of increasing mineral N release and the presence of more labile organic matter, respectively, from areas with brash. Mäkiranta et al. (2012) observed over 3-fold increase in seasonal average (over 3 years between May – Oct) soil $N_2O$ flux and 2-fold in $CO_2$ efflux from plots with logging residues than without but observed no change in $CH_4$ emissions. This indicates that brash





removal post-harvesting (e.g. for biofuel as suggested by Mäkiranta et al., 2012) might be a way of limiting GHG effluxes

from peat decomposition. More information on GHG fluxes from brash and stumps, and the underlying soil processes that

might be influenced by felling is a priority for future research.

**5 Conclusions**

In this upland Sitka spruce plantation on organic-rich peaty gley soil, clearfell harvesting affected soil GHG fluxes by

increasing soil temperature and moisture content and reducing fine root mass which affected the soil nutrient and organic C

supply and associated microbial populations, activities, and decomposition rates. Although soil moisture increased

significantly after felling, there were no direct correlations with the soil GHG fluxes probably because there was limited

variation in the high soil moisture after felling. By contrast, there was a good correlation between GHG fluxes and the soil

temperature which exhibited much larger temporal variation. This study does not take into account fluxes from brash

decomposition, because of the small flux chamber areas, and therefore our total measured soil GHG efflux after felling

probably underestimate that of the site as a whole. Over the 3-year measurement period after felling, soil $CO_2$ effluxes reduced

substantially (55%) due to cessation of root respiration outweighing increased decomposition. For the same period, $CH_4$ fluxes

changed from a small net sink to a net source, increasing throughout and $N_2O$ fluxes increased substantially in the 2 years after

felling. Mean soil $CO_2$, $CH_4$ and $N_2O$ fluxes over the 3-year period after felling contributed 83, 1 and 16%, respectively, to

total GHG flux on a $CO_2e$ basis with an overall reduction of 45%, due to much larger soil $CO_2$ flux reduction than the combined

soil $CH_4$ and $N_2O$ flux increases.

**Author contributions.** SY, JILM, GX and MP designed the study. GX, AA and SY carried out the gas sampling and soil
environmental measurements. JB carried out the GC analyses, and SY did all the data analyses. JF was responsible for
statistical analysis and SY prepared the paper, with contributions from JILM and GX.


**Competing interests.**    The authors declare that they have no conflicts of interest.

*Acknowledgements.*    We would like to thank Forest Research colleagues Elena Vanguelova, Ed Eaton, Sue Benham and
Vladimir Krivtsov for soil and vegetation sampling and analysis. We are indebted to the Forestry England staff who gave
permission for the study in Harwood Forest, and for their huge support throughout; particularly the late Jonathan Farries. We
also thank Russell Anderson for his valuable comments on the paper.

*Financial support.*    This work was supported by the Forestry Commission.

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
