# Peer review of "Effects of clearfell harvesting on soil CO2, CH4 and N2O fluxes in an upland Sitka spruce stand in England"

_Biogeosciences, 2021_

## Author Response (AR1)

**Interactive Discussion**

Associate Editor Decision: Reconsider after major revisions (16 Apr 2021) by Ben Bond-Lamberty

Comments to the Author:

General comments

Thanks for your submission, which was read by two reviewers, who both provide thoughtful comments; thanks also for your initial responses to these comments, which are in general comprehensive and considered. R1's questions about scientific significance and contextualizing the GWP results should be considered carefully—particularly their integration into the discussion—but your initial responses proposing some moderate modifications seem reasonable.

I have read the ms and broadly agree with the reviewers on the generally well-done nature of this study and its potential interest, but also find that it needs moderate to major revisions in many areas. In addition to the reviewers' comments, the methods (in particular with respect to code and data availability) need more clarity, as do the statistical tests being performed, and several figures should be re-thought or improved. See specific comments below.

In summary, this is an interesting ms with many strengths. It needs revisions in many areas for clarity, concision, and reproducibility, and will then be re-reviewed.

**Response:** We are grateful for the careful consideration of our paper by the associate editor and have considered all the points; our responses are given below:

**Specific comments**
**1. Line 22:** up to 20% of annual emissions in what sense? GWP? Clarify
**Response:** We rephrased this to make it clear it is a comparison of their GWP.

**2. Figure 1:** not a great map, as it shows no lat/lon and a lot of extraneous information
**Response:** We previously added this information about copyright on the map as requested by BG journal editorial support team. However, we've now removed this from the map and added it to the Figure caption. We also mentioned the lat/lon in the section "Site description and study layout".

**3. L. 103:** provide exact locations of study areas
**Response:** As mentioned above, we added the lat/log to the text to show the study location.

**4. L. 130-131:** citation for HRM package? And, your R version was from 2016, really? Related, in line 186 provide versions number of R and all packages used
**Response:** Thanks for pointing out the mistake in the R version. We've now added the version of the R HMR package used which was "version 1.0.1" and updated the citation and the reference to "Pedersen et al., 2020" accordingly.

**5. L. 140-150:** good, useful comparison
**Response:** Thanks

**6. Figure 2:** useful but incomplete. Report intercepts for the fitted regressions along with RMSE
**Response:** The intercepts was not significantly different from zero, so regression was through the origin; we have added this information to the caption.

As requested we also cited in the Figure caption that the Root Mean Square Error (RMSE) values were 0.37 and 0.41 g $CO_2$-C m$^{-2}$ d$^{-1}$ for area A and area B respectively.

**7. L. 172:** what function exactly? Provide more information on Q10 calculation, including temperature range

**Response**: We added an equation to show how the Q10 values were calculate and updated the reference. Temperature ranges are discussed in the Results section 3.1 and shown in Fig. 3a (now Fig. 2a).

8. Data and code: please specify availability and, ideally, deposit them in a permanent repository for scientific reproducibility and transparency

**Response**: We have added the R HMR package version that we used, updated and revised all the relevant references in the statistical analysis section including the nlme in the statistical analysis section.

All the date and the code from the statistical analysis are available on required from the corresponding author, as stated in other papers.

9. Figure 4: a cumulative line for each area might be useful?

**Response:** Thanks for the suggestion, however we've revised the Fig as line graph, as suggested by referee 1, and we think it is clear now.

10. Figure 5 and figures generally: you're using R, which does beautiful data visualizations; it seems a shame to fall back to Excel for graphs

**Response:** We agree with the editor's comment and have now redrawn all the Figures in R for consistency.

11. L. 454: start new paragraph for readability

**Response:** Thanks for the suggestion, we added a new paragraph.

12. Figure 7: colour would improve this?

**Response:** Thanks for the suggestion, we've now revised the Fig in colour as suggested.